# Lactate Reprograms Energy and Lipid Metabolism in Glucose-Deprived Oxidative Glioma Stem Cells

**DOI:** 10.3390/metabo11050325

**Published:** 2021-05-18

**Authors:** Noriaki Minami, Kazuhiro Tanaka, Takashi Sasayama, Eiji Kohmura, Hideyuki Saya, Oltea Sampetrean

**Affiliations:** 1Division of Gene Regulation, Institute for Advanced Medical Research, Keio University School of Medicine, 35 Shinanomachi, Shinjuku-ku, Tokyo 160-8582, Japan; noriakiminami@me.com (N.M.); hsaya@a5.keio.jp (H.S.); 2Department of Neurosurgery, Kobe University Graduate School of Medicine, 7-5-1, Kusunoki-cho, Chuo-ku, Kobe 650-0017, Japan; kazutana@med.kobe-u.ac.jp (K.T.); takasasa@med.kobe-u.ac.jp (T.S.); ekohmura@med.kobe-u.ac.jp (E.K.)

**Keywords:** cancer stem cell, glioma, glioma stem cell, glucose deprivation, lactate, metabolism, lipid metabolism, metabolic cooperation, metabolic reprogramming, metabolic symbiosis

## Abstract

Fast-growing tumors satisfy their bioenergetic needs by supplementing glucose with alternative carbon sources. Cancer stem cells are the most versatile and robust cells within malignant tumors. They avoid potentially lethal metabolic and other types of stress through flexible reprogramming of relevant pathways, but it has remained unclear whether alternative carbon sources are important for the maintenance of their tumor-propagating ability. Here we assessed the ability of glycolytic and oxidative murine glioma stem cells (GSCs) to grow in an ultralow glucose medium. Sphere formation assays revealed that exogenous lactate and acetate reversed the growth impairment of oxidative GSCs in such medium. Extracellular flux analysis showed that lactate supported oxygen consumption in these cells, whereas metabolomics analysis revealed that it increased the intracellular levels of tricarboxylic acid cycle intermediates, ATP, and GTP as well as increased adenylate and guanylate charge. Lactate also reversed the depletion of choline apparent in the glucose-deprived cells as well as reprogrammed phospholipid and fatty acid biosynthesis. This metabolic reprogramming was associated with a more aggressive phenotype of intracranial tumors formed by lactate-treated GSCs. Our results thus suggest that lactate is an important alternative energetic and biosynthetic substrate for oxidative GSCs, and that it sustains their growth under conditions of glucose deprivation.

## 1. Introduction

Cancer cells require large amounts of glucose to fuel their growth. Glucose availability affects the acetyl coenzyme A (CoA) pool, ATP formation, redox balance, and de novo synthesis of membrane components such as phospholipids. Fast-growing tumors such as glioblastoma and brain metastases manifest a “bioenergetic substrate gap,” in that a substantial portion of the intracellular acetyl-CoA pool is not derived from circulating glucose [1,2]. The ability of tumors to reprogram their metabolism in order to allow the use of additional or alternative substrates such as acetate [2], lactate [3], or fatty acids [4,5,6] and thereby to close such energy gaps determines their survival under conditions of metabolic stress.

Cancer stem cells (CSCs) are the most versatile and robust cells within malignant tumors. They avoid potentially lethal conditions such as metabolic stress by reprogramming relevant pathways. Their metabolism is highly altered, with the changes being specific to the original tumor type [7,8,9,10,11]. Furthermore, stem cell–like populations such as those in breast cancer and glioblastoma also manifest metabolic plasticity [12,13], a feature that can lead to the emergence of therapy-resistant phenotypes. Metabolic flexibility of CSCs has been linked to oncogene expression, specific transcription factors, and reprogramming in response to hypoxia [12,13,14]. However, it has remained unclear whether the closing of energy gaps through the adoption of alternative carbon sources is also important for the growth of CSCs and CSC-based tumors.

Lactate has emerged as a major circulating substrate of the tricarboxylic acid (TCA) cycle that is utilized by both normal tissue and tumors [3,15,16,17]. It can achieve a concentration of 2 mM in blood under physiological conditions [18] and can be taken up by muscle and brain tissue [16]. In the brain, it is also released by astrocytes and consumed by neurons, a phenomenon known as the astrocyte-neuron lactate shuttle [19,20,21]. In neurons, lactate serves as a substrate for oxidative metabolism and is able to support cell survival for up to 24 h even in the absence of glucose [22]. Lactate also contributes to energy generation in genetically engineered lung and pancreatic tumors in fasted mice [15]. Moreover, a lactate shuttle has been found to operate in tumors, with glycolytic and oxidative tumor cells mutually adjusting their utilization of energy metabolites [23].

We previously showed that glioma stem cells (GSCs) in a mouse model of glioblastoma contain subpopulations that are predominantly glycolytic and release lactate (glycolytic GSCs), as well as those that rely largely on oxidative phosphorylation (oxidative GSCs) [12]. We have examined here whether GSCs are able to use lactate as an alternative carbon source and how such use might affect their metabolism. We found that lactate is able to serve as a major energetic substrate in glucose-deprived oxidative GSCs. In addition to restoring TCA intermediates, lactate modulated redox status, reprogrammed lipid metabolism, and rescued the growth of GSCs and GSC-based tumors.

## 2. Results

### 2.1. Lactate Sustains the Growth of Oxidative GSCs during Glucose Deprivation

To investigate whether lactate can serve as an additional bioenergetic substrate for GSCs, we first determined the glucose concentration that allowed mouse GSCs to remain viable but unable to proliferate or form spheres. We found this glucose concentration to be 0.175 mM, which corresponds to 1% of the concentration in our standard culture condition. We have previously shown that GSCs can differ in their main energy pathway, and have established two clonal GSC populations, glycolytic GSCs (GSC^gly^, line GSCA), and oxidative GSCs (GSC^mit^, line GSCB) [12], as well as a non-clonal population containing predominantly oxidative GSCs (TSH) [12,24]. Here, we first cultured GSCA and GSCB in in a medium containing 0.175 mM glucose (ultralow glucose medium, or ULGM) and various concentrations of lactate (Figure 1A). The rescue effect of lactate on cell growth was small in glycolytic GSCs, even at a concentration of 4 mM (Figure 1B). In contrast, lactate was consumed by oxidative GSCs (Figure 1C) and sustained their growth as spheres (Figure 1D and Appendix A). Acetate has also been identified as a bioenergetic substrate for glioma cells [2,25]. We found that the addition of 4 mM acetate had a significant rescue effect on the growth of oxidative GSC spheres, but this effect was not as prominent as that of equimolar lactate (Figure 1E). Pyruvate is taken up by cells through the same monocarboxylate transporters (MCTs) as is lactate, and it has been shown to serve as an alternative fuel in metastatic breast cancer cells and cultured glioblastoma cell lines [26,27]. We found that the addition of pyruvate to oxidative GSCs did not, however, rescue sphere growth. Indeed, pyruvate had a slight inhibitory effect on GSC growth, as did oxaloacetate (Figure 1F,G). The TCA intermediates citrate, α-ketoglutarate, succinate, fumarate, and malate had no significant effect on sphere growth (Figure 1H).

### 2.2. Expression of LDHB Correlates with the Rescue Effect of Lactate in Glucose-Deprived GSCs

We next investigated the molecular basis of the effect of lactate on sphere growth. The difference in lactate utilization between oxidative versus glycolytic GSCs might have reflected a difference in lactate uptake or intracellular fate (Figure 2A). Consistent with their respective propensities to release [12] or consume (Figure 1B–D) lactate, immunoblot analysis revealed that glycolytic GSCs preferentially expressed MCT4 whereas oxidative GSCs preferentially expressed MCT2 (Figure 2B). Lactate dehydrogenase isozyme B (LDHB) catalyzes the conversion of lactate to pyruvate (Figure 2A), which is necessary to route lactate into subsequent catabolic reactions. Therefore, next we analyzed the expression of LDH isoforms in the glycolytic GSC line (GSCA) and the two oxidative GSC lines (GSCB and TSH). The expression of LDHA was markedly higher in the lactate-producing GSCA cells than in GSCB or TSH cells, whereas LDHB was more abundant in the lactate-consuming GSCB and TSH cells (Figure 2C). To further examine the relation between LDHB expression and lactate utilization, we performed single-cell cloning for the TSH cells and established 10 clonal populations, designated TSH-N1 to TSH-N10. The contribution of lactate to sphere growth was quantified as the lactate-induced change in sphere area for each clone (Figure 2D) and was then plotted against the abundance of *Ldha* or *Ldhb* transcripts (Figure 2E,F). A Pearson coefficient of 0.675 was suggestive of a correlation between *Ldhb* expression and lactate use (Figure 2F). On the basis of these results, we selected TSH-N5 as a representative clonal line for subsequent experiments.

### 2.3. Lactate Supports Energy Production and Modulates Redox Balance in Glucose-Deprived GSCs

The rescue effect of lactate on sphere growth was absent under hypoxic conditions (Figure 3A), suggesting that the lactate-induced proliferation of GSCs is supported by oxygen-dependent metabolism. One fate of lactate under aerobic conditions is to enter the TCA cycle and thereby to contribute to energy production as an oxidation substrate [3,23]. Indeed, lactate increased oxygen consumption (Figure 3B) and ATP content (Figure 3C) in glucose-deprived TSH-N5 cells, suggestive of a contribution to energy production. The increase in oxygen consumption induced by addition of 4 mM lactate for 1 h was reversed by addition of MCT inhibitor α-cyano-4-hydroxycinnamate (α-CHCA) [23], in a concentration-dependent manner (Appendix A). Lactate also increased the intracellular levels of glutathione (GSH) (Figure 3D) and total NADPH plus NADP^+^ (Figure 3E). In contrast, the level of reactive oxygen species (ROS) was significantly reduced by lactate exposure (Figure 3F). Together, these results suggested that lactate serves as both an oxidation substrate and a modulator of redox balance in GSCs.

### 2.4. Lactate Reprograms Catabolic Processes in Glucose-Deprived GSCs

To investigate the extent of the metabolic changes induced by prolonged utilization of lactate in glucose-deprived GSCs, we measured the levels of 116 core intracellular metabolites by capillary electrophoresis (CE) coupled with mass spectrometry (MS) in TSH-N5 cells cultured in ULGM or ULGM supplemented with 4 mM lactate for 6 days. Consistent with the bioenergetic changes apparent after 24 to 72 h (Figure 3B,C), exposure of the cells to lactate for 6 days increased the levels of ATP and GTP as well as the total adenylate or guanylate charge (Figure 4A). Lactate also increased the intracellular levels of citric acid, fumaric acid, and malic acid (Figure 4B,C). Unexpectedly, the VIP (variable importance in projection) scores as determined by partial least squares–discriminant analysis (PLS-DA) were highest for choline and glycerol 3-phosphate (Figure 4C,D), suggestive of a contribution of lactate to lipid synthesis. The NAD^+^ level was also increased in ULGM cells exposed to lactate (Figure 4E). Furthermore, metabolite set enrichment analysis identified phospholipid biosynthesis as the top differential set (Figure 4F). Together, these results showed that lactate is able to induce reprogramming of catabolic processes and to serve as an alternative bioenergetic substrate in GSCs, and they suggested that it might also make a previously unrecognized contribution to anabolic processes.

### 2.5. Lactate Reprograms Lipid Metabolism in Glucose-Deprived GSCs

To further investigate the effect of lactate on lipid metabolism, we measured relevant metabolites by liquid chromatography (LC) and MS. Principal component analysis of lipids from GSCs cultured in ULGM with or without 4 mM lactate for 6 days showed a marked separation between the two conditions (Figure 5A). Metabolite set enrichment analysis revealed that phospholipid biosynthesis and fatty acid metabolism were the top significantly different sets (Figure 5B). 1,2-Distearoyl-glycero-3-phosphocholine and lyso-phosphoglycerol (18:1), both of which are intermediates of phospholipid synthesis, as well as palmitoylcarnitine, which is related to fatty acid metabolism, were the metabolites most significantly enriched in the ULGM+L group (Figure 5C, Appendix A).

### 2.6. Lactate Sustains Aggressiveness of GSCs during Glucose Deprivation

Finally, we tested whether GSC-derived tumors are influenced by the metabolic changes induced by utilization of lactate as an alternative fuel. To assess whether metabolic reprogramming can occur in tumors formed by GSCs in the presence of abundant glucose, we implanted GSCB cells into the right forebrain of immunocompetent mice and allowed them to form tumors. We then prepared coronal brain slices and cultured them in ULGM or ULGM supplemented with 4 mM lactate. Tumors in slices cultured in ULGM supplemented with lactate showed a tendency to grow compared with those in slices cultured in ULGM alone, but this difference failed to achieve statistical significance (Figure 6A,B). In contrast, when GSCB cells were first cultured in ULGM with or without lactate for 6 days and then implanted subcutaneously into immunodeficient mice or intracranially into immunocompetent mice (Figure 6C), tumors formed by the lactate-treated GSCs were significantly more aggressive than were those formed by GSCs cultured in ULGM alone (Figure 6D,E).

## 3. Discussion

Supplementation of carbon sources by fuels other than glucose allows rapidly dividing malignant cells to meet their energetic needs. In cancer cells with stemlike properties, which are able to adapt to and survive under adverse conditions, such supplementation might result in metabolic reprogramming and the emergence of stress-resistant phenotypes. We asked here whether GSCs can utilize alternative substrates, and we found that lactate is able to sustain both energy and biomass metabolism and thereby support the growth of these cells.

A key finding of the present study is that exogenous lactate allowed oxidative GSCs to grow as spheres in the presence of a glucose concentration that was able to sustain cell viability but not cell proliferation. The increase in oxygen consumption rate and intracellular ATP content induced by such lactate supplementation suggested that lactate served as a TCA substrate in GSCs maintained under aerobic conditions. The use of lactate as an energetic fuel has previously been described for several cancer cell lines in culture [23,28], for cancer stem cells in patient-derived colorectal cancer organoids [29], and for lung tumors in vivo [3,15]. Our results now extend these findings to GSCs, with implications for understanding of malignant glioma metabolism.

The GSCs used in our study were established by transformation of neural stem/progenitor cells with the oncoprotein H-Ras^V12^ [12,24,30]. Given that neuronal stem cells are able to survive on lactate for up to 24 h [22], the ability of GSCs to use lactate might reflect a characteristic of their cell of origin. However, we previously found that GSCs with the same genetic background and established by the same transformation process rely on either mitochondrial or glycolytic metabolism [12,31]. Of these two types of GSC, only the former was able to utilize lactate as a fuel in the present study. These oxidative GSCs showed preferential expression of LDHB, which catalyzes the conversion of lactate to pyruvate, whereas glycolytic GSCs showed preferential expression of LDHA, which catalyzes the reverse reaction. Of the two LDH isoforms, only the abundance of transcripts for LDHB showed a significant correlation with lactate-fueled growth. Given that inhibition of LDHA activity has been found to induce differentiation and death of glycolytic GSCs [32], LDH isoforms might be important metabolic regulators of and therapeutic targets for these two types of GSC. However, we previously found that the abundance of LDHA rapidly fluctuates in GSCs in response to hypoxia [12], suggesting that the individual isozymes might be markers rather than key regulators of the metabolic profile of GSCs in a given environment.

Although only addressed briefly in the present study, the tumor microenvironment is a major determinant of the use of exogenous lactate. We found that oxygen at a concentration of more than 1% was required for lactate-dependent sphere growth of glucose-deprived GSCs. Lactate use in cancer cell lines was also observed under normoxic culture conditions [23,28]. In addition, evidence for its use by tumors in vivo has come from lung tumors, which exist in an aerobic environment. In genetically engineered models of lung and pancreatic tumors, the former thus relied on lactate, whereas the latter, well known to be hypoxic, relied more on glutamine as an alternative fuel [15]. The brain is also an aerobic environment, with some GSCs having been found to reside in a well-oxygenated perivascular niche [33]. Astrocytes, differentiated glioma cells, and glycolytic GCSs all compete for glucose and release lactate, which might be expected to generate a local environment with sufficient oxygen but insufficient glucose (Figure 7). Metabolic cooperation between lactate-producing non-CSCs and lactate-consuming CSCs has been observed in colorectal cancer [29]. In the case of malignant glioma, glycolytic GSCs and oxidative GSCs might also engage in such a symbiotic relationship, which might complicate targeting of GSCs by metabolic inhibitors.

A second key finding of our study is that lactate consumption by GSCs resulted in a significant decrease in ROS abundance in association with a significant increase in GSH and total NADPH plus NADP^+^ levels. Cellular redox status reflects the constantly changing balance between pro- and antioxidants. It also differs among subcellular compartments and cell types. Whereas the contribution of lactate to individual pro- and antioxidant factors remains unclear, the presence of lactate as an alternative energy source might reduce oxidative stress by allowing the trace amounts of glucose to be rerouted to pathways that produce reducing agents, such as the pentose phosphate pathway. Alternatively, the influx of electrons that accompanies lactate uptake, and/or the reprogramming of energy and central carbon metabolism itself, might lead to a net increase in reducing agents. Indeed, the fact that lactate but not equimolar pyruvate, which can also contribute to TCA cycle metabolites and ATP production in glucose-deprived glioma cells [27], rescued sphere growth could be partially due to more reducing equivalent NADH being produced during lactate versus pyruvate utilization. Whereas such hypotheses need to be investigated further in future studies, our present data suggest that the use of lactate as an alternative carbon source in glucose-deprived GSCs supports cell proliferation not only through the recovery of energy status, but also through the modulation of redox balance.

A similar argument can be made for the changes in lipid metabolism apparent in lactate-consuming GSCs. The fact that these cells can proliferate for >1 week is indicative of active membrane formation. Indeed, metabolomics analysis revealed a significant difference in phospholipid biosynthesis and fatty acid metabolism between glucose-deprived oxidative GSCs cultured in the absence or presence of lactate. As with redox balance, this difference may reflect the rerouting of available glucose in the lactate-consuming cells. However, the tracing of ^13^C-labeled lactate in HeLa and H460 cells revealed a direct contribution of lactate to the lipid pool [28]. Furthermore, lactate also contributed to acetyl-CoA pools in neurons and subsequent lipid droplet accumulation in glial cells in Drosophila [21]. We were not able to distinguish between these two scenarios in the present study, although the observed rescue of the complete depletion of choline together with the significant increase in citrate abundance induced by lactate in the glucose-deprived cells is suggestive of a direct contribution.

The most important finding of the present study, however, is that the metabolic changes induced by lactate in GSCs are sufficiently robust to support the tumor-propagating potential of these cells. Exposure of GSCs to lactate as an alternative carbon source for 6–7 days thus not only rescued sphere growth in vitro, but also resulted in a significant decrease in the survival of mice harboring tumors formed by the cells in vivo. Given that the rapid growth of glioblastoma generates a heterogeneous environment including areas of restricted glucose availability, and that other substrates such as acetate and cholesterol are also available in the brain, it is possible that even a transient exposure to such conditions might induce a marked change in the phenotype of GSCs. Our results thus suggest that GSCs should be considered as an evolving therapeutic target, and that a continued search for vulnerabilities of these cells least affected by metabolic reprogramming is warranted.

## 4. Materials and Methods

### 4.1. GSCs and Cell Culture

The clonal GSC lines GSCA and GSCB were derived from murine *Ink4a/Arf*-null neural stem/progenitor cells expressing the fluorescent protein ds-Red and H-Ras^V12^ (glioma-initiating cells, GIC-A and GIC-B) and were established as previously described [12,31]. TSH is a stem cell line derived after one passage in vivo from GIC-H cells (sorted by hygromycin selection) [12,24]. TSH-N1 to TSH-N10 cells were established by single-cell cloning from the TSH line. All cells were maintained in serum-free neural stem cell medium (NSM), as previously described [30]. We determined that a glucose concentration of 0.175 mM, which corresponds to 1% of the standard concentration, allowed GSCs to remain viable but not to proliferate or form spheres. A modified NSM was prepared from Dulbecco’s modified Eagle’s medium (DMEM)–F12 lacking glucose, pyruvate (Research Institute for Functional Peptides, Yamagata, Japan), and by supplementation with concentrations of these chemicals as indicated in each experiment. The modified NSM with a glucose concentration of 0.175 mM was designated as the ultralow glucose medium (ULGM). For experiments with cells maintained in ULGM, the medium was replenished daily to avoid complete glucose starvation. All media was adjusted to pH 7.4 before use. All cells were studied within 20 passages after establishment.

### 4.2. Sphere Growth Assay

Cells were plated in ultralow attachment 96-well plates (Corning, Corning, NY, USA) at a density of 100 (normoxia) or 1000 (hypoxia) cells per well, and they were cultured in ULGM or ULGM supplemented with alternative substrates, as indicated. The outermost wells of each plate were excluded from analysis to avoid artifacts due to medium evaporation. Images were acquired with a Cell^3^ iMager Duos (SCREEN Holdings, Kyoto, Japan) or BZ-X710 (Keyence, Osaka, Japan) inverted microscope at 7 days after plating. Sphere area was quantified with Cell^3^ iMager Duos Software v1.4 (SCREEN Holdings, Kyoto, Japan) or ImageJ software ver1.51s (NIH, Bethesda, MD, USA). For alternative substrate supplementation, L-lactic acid was purchased from Wako (Osaka, Japan), and all other chemicals from Tokyo Chemical Industry (Tokyo, Japan).

### 4.3. Measurement of Lactate Concentration

Cells were plated at a density of 1 × 10^6^ per well in six-well plates and cultured in ULGM supplemented with 4 mM lactate for up to 96 h without medium replenishment. The lactate concentration in the medium was measured every 24 h with the use of a Lactate Pro 2 Assay Kit (Arkray, Kyoto, Japan).

### 4.4. Immunoblot Analysis

Immunoblot analysis was performed according to standard protocols, as previously described [31]. In brief, cells were washed with PBS and lysed in radioimmunoprecipitation buffer. The resulting lysates were fractionated by SDS-polyacrylamide gel electrophoresis, and the separated proteins were transferred to a polyvinylidene difluoride membrane (Bio-Rad, Hercules, CA, USA) and exposed to primary antibodies to MCT2 (Santa Cruz Biotechnology, Dallas, TX, USA, sc-166925), MCT4 (Santa Cruz Biotechnology, Dallas, TX, USA, sc-376140), LDHA (Thermo Fisher Scientific, Rockford, IL, USA, PA5-27406), LDHB (Abcam, Cambridge, UK, ab85319), and to α-tubulin (Santa Cruz Biotechnology, Dallas, TX, USA, sc-32293). Immune complexes were detected with HRP-conjugated secondary antibodies and Chemiluminescence Reagent (PerkinElmer, Waltham, MA, USA).

### 4.5. RT and Real-Time PCR Analysis

Total RNA was extracted from cells with the use of an RNeasy Mini Kit (Qiagen, Hilden, Germany), and portions (500 ng) of the RNA were subjected to reverse transcription (RT) using the Prime Script RT Reagent Kit with gDNA Eraser (Takara Bio, Shiga, Japan). Real-time polymerase chain reaction (PCR) analysis was performed with TB Green Premix Ex Taq (Takara Bio) in a Thermal Cycler Dice Real Time System (TP800; Takara, Shiga, Japan). The amplification protocol was as follows: Initial denaturation, 95 °C for 30 s, 40 cycles of 95 °C for 5 s and 60 °C for 30 s. The threshold cycle value was calculated by the second-derivative maximum method (Ct-SDM, Takara Bio), and the abundance of each target mRNA was normalized by that of *Actb* mRNA. The PCR primers (forward and reverse, respectively) were as follows: *Actb*, 5′-CATCCGTAAAGACCTCTATGCCAAC-3′ and 5′-ATGGAGCCACCGATCCACA-3′; *Ldha*, 5′-GAACTGGGCACTGACGCAGA-3′ and 5′-CCAATGGCCCAGGATGTGTA-3′; and *Ldhb*, 5′-AAGTACAGCCCTGACTGCACCA-3′ and 5′-TTGCATCCGCTTCCAATCAC-3′.

### 4.6. Measurement of OCR

Oxygen consumption rate (OCR) was measured with a Seahorse XF Extracellular Flux Analyzer (Agilent Technologies, Santa Clara, CA, USA). Cell plating and measurements were performed as previously described [31]. In brief, cells were incubated in ULGM, ULGM with 4 mM lactate for 24 h. The medium was then changed to glutamine-free Seahorse XF base medium (Agilent Technologies, Santa Clara, CA, USA) supplemented with 0.175 mM glucose with or without 4 mM lactate, and OCR was measured every 10 min for 100 min. The basal OCR recorded 60 min after the initial measurement was compared between groups.

### 4.7. Measurement of Intracellular ATP

Cells were incubated for 72 h in ULGM or ULGM with 4 mM lactate, dissociated, transferred to 96-well plates at a density of 1 × 10^4^ per well, and cultured for 3 h in the respective medium. ATP content of the cells was then measured with the use of Cell Titer-Glo Assay Kit (Promega, Madison, WI, USA).

### 4.8. Measurement of Total NADPH Plus NADP^+^ and GSH Levels

Cells were cultured in ULGM or ULGM with 4 mM lactate for 72 h, transferred to 96-well plates at a density of 5 × 10^4^ per well (NADPH plus NADP^+^ assay) or 2.5 × 10^4^ per well (GSH assay) in the corresponding medium, and incubated for 72 h. Total NADPH plus NADP^+^ or GSH levels were then measured with the use of an NADP/NADPH-Glo Assay Kit (Promega, Madison, WI, USA) or GSH-Glo Assay Kit (Promega, Madison, WI, USA). All measurements were performed with an EnVision microplate reader (PerkinElmer, Waltham, MA, USA).

### 4.9. Measurement of ROS

Cells were incubated for 72 h in ULGM or ULGM supplemented with 4 mM lactate and were then exposed for 30 min to 5 mM CellROX deep red (Thermo Fisher Scientific, Waltham, MA, USA). The fluorescence intensity of the dissociated cells was measured with an Attune flow cytometer (Thermo Fisher Scientific, Waltham, MA, USA) and analyzed with FlowJo software (Tree Star Inc, San Carlos, CA, USA).

### 4.10. Metabolomics Analysis

For extraction of metabolites, cells were mechanically dissociated, washed with 5% mannitol, and treated with methanol (for CE) or ethanol (for LC) and internal standards. Extracts for CE and time-of-flight MS (TOFMS) and for CE and tandem MS (MS/MS) were filtered by centrifugation in an Ultrafree MC-PLHCC filter unit (Human Metabolome Technologies, Tsuruoka, Yamagata, Japan), whereas those for LC-TOFMS were subjected to sonication for 5 min and then centrifuged to remove debris. All samples were stored at –80 °C until analysis. Quantitative analysis of metabolites was performed by Human Metabolome Technologies (Tsuruoka, Yamagata, Japan) based on methods previously described [34,35,36,37,38]. For analysis of 116 core metabolites, CE-TOFMS (cation analysis) and CE-MS/MS (anion analysis) were carried out with an Agilent CE system equipped with an Agilent 6210 TOF mass spectrometer (Agilent Technologies, Santa Clara, CA, USA) or an Agilent 6460 Triple Quadrupole LC/MS system (Agilent Technologies, Santa Clara, CA, USA), respectively. The TOF mass spectrometer was scanned from a mass/charge ratio (*m*/*z*) of 50 to 1000, and the triple-quadrupole mass spectrometer was set to the dynamic MRM mode. For analysis of lipids, LC-TOFMS analysis was carried out with the *LC-TOFMS* package of HMT, with the use of an Agilent 1200 HPLC pump equipped with an Agilent 6210 TOF mass spectrometer. Peaks were extracted with MasterHands version 2.17.4.19 (Keio University, Tsuruoka, Yamagata, Japan) [36] and MassHunter Quantitative Analysis B.04.00 (Agilent Technologies, Santa Clara, CA, USA) in order to obtain peak information including *m*/*z*, peak area, and migration time. For LC-TOFMS, 57 peaks were extracted and annotated. Relative levels of the analyzed metabolites were obtained by normalizing areas of the annotated peaks based on internal standard levels and sample amounts (Appendix A). Metabolite levels were normalized by cell number determined from a parallel sample.

### 4.11. Animal Experiments

Six-week-old female C57/B6J mice were injected intracranially as previously described [30] with 1 × 10^3^ GSCB cells that had been cultured for 6 days in ULGM with or without 4 mM lactate. BALB/c nu/nu mice were injected subcutaneously in the right flank with 1 × 10^5^ GSCB cells that had been similarly cultured in ULGM with or without 4 mM lactate. Animals were monitored daily and were killed when they developed neurological deficits or became moribund.

### 4.12. Confocal Microscopy of Brain Explants

Tumor-containing brain slices were established from mice injected intracranially with 1 × 10^3^ GSCB cells. Explants were placed on Millicell-CM culture inserts (Merck Millipore, Billerica, MA, USA) in glass-bottom dishes as previously described [39] and cultured in ULGM or ULGM supplemented with 4 mM lactate for 4 days. Images were acquired with an FV10i inverted confocal laser microscope (Olympus, Tokyo, Japan) on day 0 and day 3. Tumor area was quantified with the use of ImageJ software [39].

### 4.13. Statistical Analysis

All experiments were performed with at least three independent biological replicates, unless otherwise stated. Data were analyzed with the tests described in figure legends with the use of GraphPad Prism (GraphPad Software, San Diego, CA, USA). PLS-DA, PC, and metabolic set enrichment analysis were performed with the use of MetaboAnalyst 4.0 software (https://www.metaboanalyst.ca, access dates: 7 January 2021, 19 January 2021, 21 January 2021) [40,41]. A *p* value of <0.05 was considered statistically significant.

## Figures and Tables

**Figure 1 metabolites-11-00325-f001:**
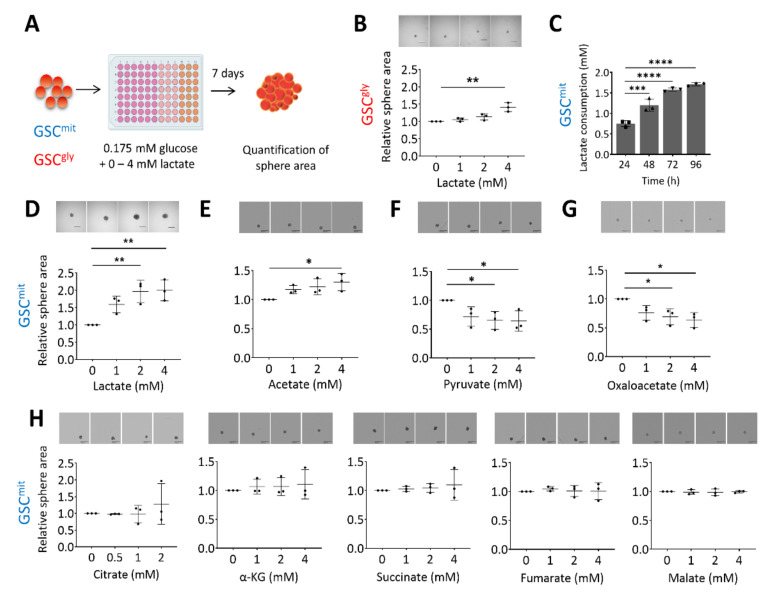
Lactate sustains the growth of oxidative GSCs during glucose deprivation. (**A**) Schematic representation of the experimental protocol. (**B**) Relative sphere growth for GSC^gly^ (GSCA) cells cultured for 7 days in ultralow glucose medium (ULGM) supplemented with the indicated concentrations of lactate. Representative images of the spheres formed in each condition are also presented. Scale bars, 500 µm. (**C**) Lactate consumption by GSC^mit^ (GSCB) cells during culture for 96 h in ULGM containing 4 mM lactate. (**D**–**H**) Relative sphere growth for GSC^mit^ cells cultured for 7 days in ULGM supplemented with the indicated concentrations of lactate (**D**), acetate (**E**), pyruvate (**F**), oxaloacetate (**G**), or citrate, α-ketoglutarate (α-KG), succinate, fumarate, or malate (**H**). All data are means ± SD from three independent experiments and were evaluated by one-way analysis of variance (ANOVA) followed by Dunnett’s post hoc test. * *p* < 0.05, ** *p* < 0.01, *** *p* < 0.001, **** *p* < 0.0001. Scheme created with BioRender.com.

**Figure 2 metabolites-11-00325-f002:**
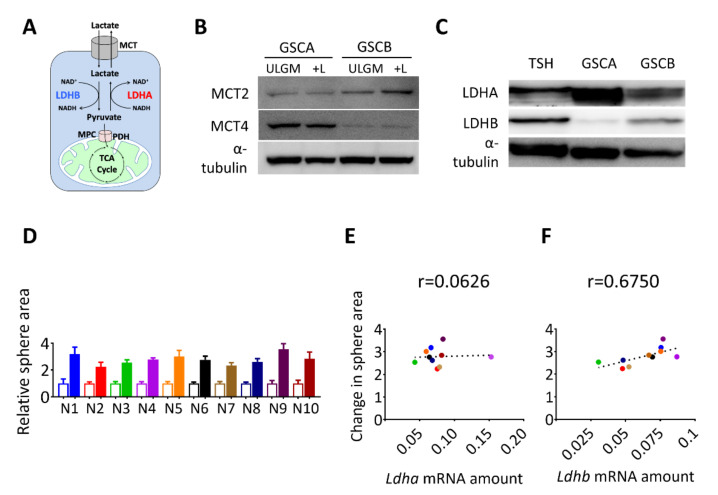
Expression of lactate dehydrogenase B (LDHB) correlates with the rescue effect of lactate in glucose-deprived GSCs. (**A**) Schematic representation of lactate uptake and lactate-pyruvate interconversion. (**B**) Immunoblot analysis of monocarboxylate transporter 2 (MCT2), MCT4, and α-tubulin (loading control) in GSC cell lines (GSCA and GSCB) cultured in ULGM or ULGM supplemented with 4 mM lactate (+L). (**C**) Immunoblot analysis of LDHA and LDHB in GSC cell lines (GSCA, GSCB, and TSH) cultured in normal medium. (**D**) Relative sphere area for TSH-derived clones (N1–N10) cultured for 7 days in ULGM (open bars) or ULGM supplemented with 4 mM lactate (filled bars). (**E**,**F**) Correlation plots of *Ldha* (**E**) or *Ldhb* (**F**) mRNA levels and lactate-induced sphere growth determined as in (**D**) for TSH clones N1 to N10. Quantitative data are means ± SD for 10-12 spheres from one representative experiment and were subjected to Pearson correlation (*r*) analysis.

**Figure 3 metabolites-11-00325-f003:**
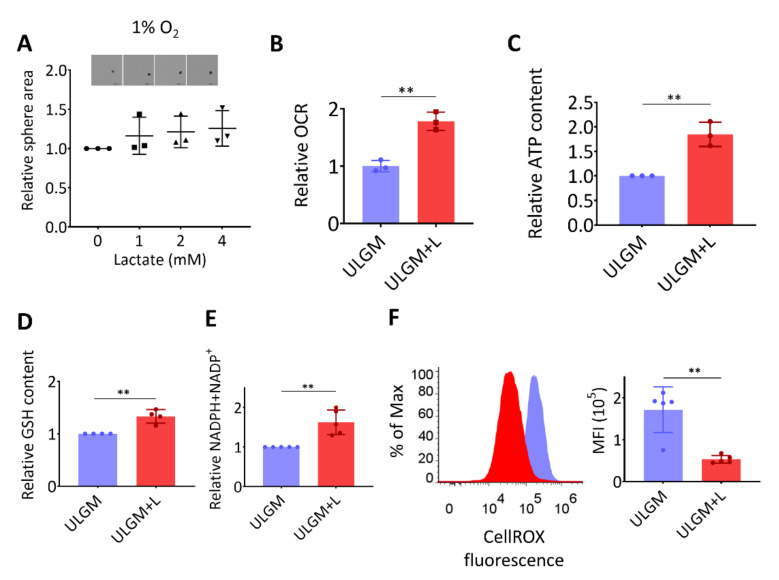
Lactate supports energy production and modulates redox balance in glucose-deprived GSCs. (**A**) Relative sphere growth for GSC^mit^ (GSCB) cells cultured for 7 days under hypoxic conditions (1% O_2_) in ULGM supplemented with the indicated concentrations of lactate. Representative images of the spheres formed in each condition are also presented. Scale bars, 500 µm. (**B**–**F**) Relative oxygen consumption rate (OCR) (**B**), intracellular ATP content (**C**), glutathione (GSH) content (**D**), and total NADPH plus NADP^+^ content (**E**) as well as representative plots and mean fluorescence intensity (MFI) of CellROX (**F**) for TSH-N5 cells cultured in ULGM or ULGM supplemented with 4 mM lactate (ULGM+L). All data are means ± SD from three to five independent experiments and were analyzed by one-way ANOVA followed by Dunnett’s post hoc test (**A**) or with the unpaired two-tailed Student’s *t* test (B–F). ** *p* < 0.01.

**Figure 4 metabolites-11-00325-f004:**
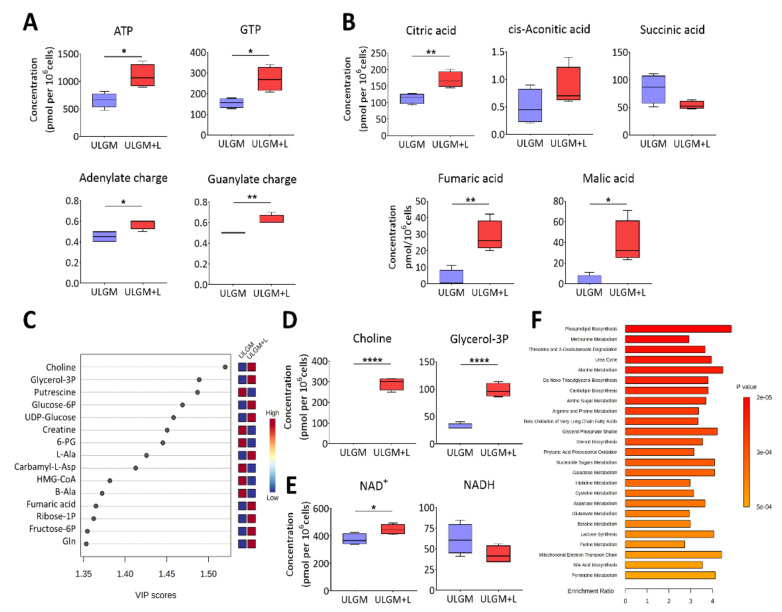
Lactate reprograms catabolic processes in glucose-deprived GSCs. (**A**) Metabolomics analysis of ATP and GTP as well as adenylate charge and guanylate charge in TSH-N5 cells cultured for 6 days in ULGM with or without 4 mM lactate. Data are presented as box-and-whisker plots, in which the boxes indicate the median and upper and lower quartile values, and the whiskers represent the highest and lowest nonoutlier values. (**B**) Concentrations of TCA cycle intermediates determined as in (**A**). (**C**) VIP (variable importance in projection) scores as determined by partial least squares–discriminant analysis (PLS-DA) for metabolites in cells as in (**A**). (**D**,**E**) Levels of choline and glycerol 3-phosphate (**D**) as well as of NAD^+^ and NADH (**E**) for cells as in (**A**). (**F**) Top 25 metabolite sets that differed significantly between ULGM and ULGM+L and their enrichment ratios. Metabolite levels were determined by capillary electrophoresis and mass spectrometry, normalized by cell number, and are from four biologically distinct replicates per condition. For PLS-DA and Metabolite Set Enrichment Analysis, data were scaled by mean-centering followed by division by the SD for each variable. * *p* < 0.05, ** *p* < 0.01, **** *p* < 0.0001 (unpaired two-tailed Student’s *t* test).

**Figure 5 metabolites-11-00325-f005:**
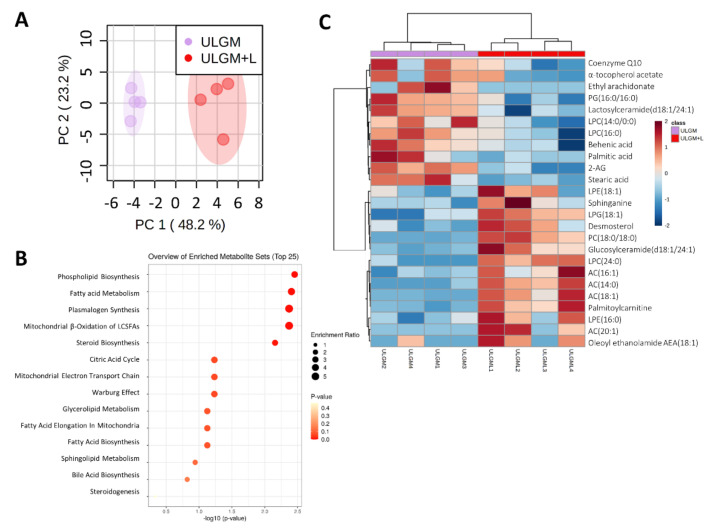
Lactate reprograms lipid metabolism in glucose-deprived GSCs. (**A**) Principal component (PC) analysis of lipid metabolite levels measured by liquid chromatography and mass spectrometry in TSH-N5 cells cultured for 6 days in ULGM with or without 4 mM lactate. (**B**) Metabolite enrichment analysis for TSH-N5 cells as in (**A**). (**C**) Heat map for the top 25 significantly different metabolites measured as in (**A**). Metabolite levels were quantified for four biological samples for each condition. Data were scaled by mean-centering followed by division by the SD for each variable. Differences between groups in (**C**) were analyzed with the unpaired two-tailed Student’s *t* test.

**Figure 6 metabolites-11-00325-f006:**
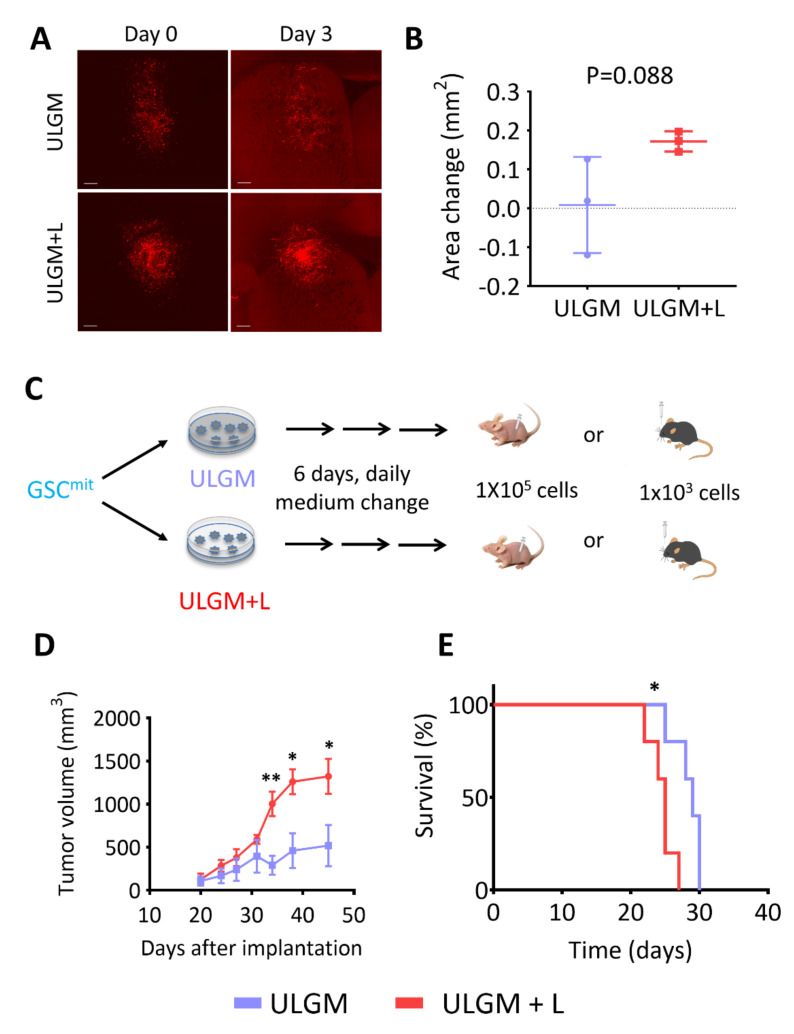
Lactate sustains aggressiveness of GSCs during glucose deprivation. (**A**) Red fluorescence images of organotypic brain explants isolated from mice harboring GSCB cell–derived tumors and cultured in ULGM or ULGM+L for 0 or 3 days. Scale bars, 300 µm. (**B**) Change in tumor area between days 0 and 3 determined as in (**A**). Data are means ± SD for three explants. The p value was determined with the unpaired two-tailed Student’s *t* test. (**C**) Schematic representation of subcutaneous or intracranial implantation of GSCB cells after maintenance in ULGM or ULGM+L for 6 days. (**D**) Tumor volume at the indicated times after subcutaneous implantation of GSCB cells as in (**C**). Data are means ± SD (*n* = 4 mice per group). * *p* < 0.05, ** *p* < 0.01 versus the corresponding value for ULGM (unpaired two-tailed Student’s *t* test). (**E**) Survival curves for mice (*n* = 5 per group) after intracranial implantation of GSCB cells as in (**C**). * *p* < 0.05 log-rank test.

**Figure 7 metabolites-11-00325-f007:**
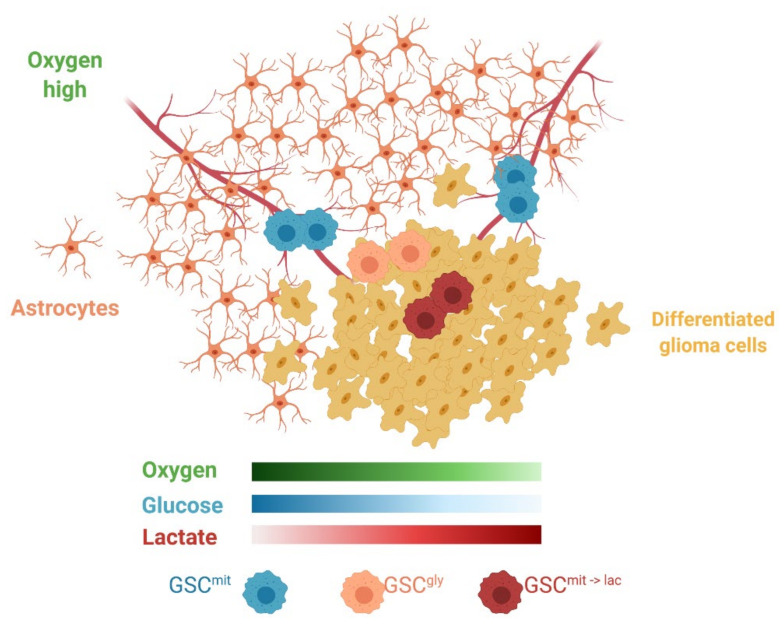
Schematic representation of the heterogeneous microenvironment present in GBM. Astrocytes, GSC^gly^, and differentiated glioma cells all consume glucose, rely on glycolysis, and release lactate, thus potentially creating areas favorable for GSC^mit^ to switch to lactate as an alternative carbon source (GSC^mit->lac^) and leading to metabolic cooperation or symbiosis. Created with BioRender.com.

## Data Availability

Data collected and analyzed in the present study are available from the corresponding author upon a request.

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
