# Peer review of "Lactate Reprograms Energy and Lipid Metabolism in Glucose-Deprived Oxidative Glioma Stem Cells"

_metabolites, 2021, doi:10.3390/metabo11050325_

Round 1

Reviewer 1 Report

The study entitled “Lactate reprograms energy and lipid metabolism in glucose-deprived oxidative glioma stem cells (GSCs)” aims to investigate the role of lactate as an alternative carbon source mediating the tumorigenicity of glioma stem cells under glucose deprivation. The authors used the sphere growth assay to assess the responses of GSCs to metabolite treatments in vitro. They also performed targeted metabolomics to examine how lactate treatment affects catabolic pathways and lipid metabolism in one of the glucose-deprived GSC lines. The tumorigenicity of the lactate-treated GSCs in vivo was determined based on tumor sizes in a subcutaneous transplantation model and survival curve from an intracranial implantation experiment. The authors conclude that lactate is a crucial substrate that sustains the tumorigenesis of GSCs under glucose-deprivation. Overall, the study presents an interesting question regarding the potentially unique metabolic dependencies of the GSCs. However, the experiment design and data from the study are not sufficient to support this view.

Major

  1. The authors need to describe better and characterize the glioma stem cells (GSCs). Specifically, how GSCs are different from their “non-stem cell” counterparts in lactate metabolism and tumorigenic potential?
  2. The implantation experiments are very appreciated. It will strengthen the study if the authors can examine the key metabolic features of the engrafted tumors.
  3. Small molecule inhibitors that block lactate metabolism should be incorporated in the study to strengthen the hypothesis.   
  4. It is unclear how TSH-N5 was generated and why this line was chosen for the subset of experiments. This information is important for the reviewer to determine if data interpretation regarding “cancer stem cell” in relation to tumorigenesis.

Minor

  1. For all the in vitro experiments, the N should be at least five.
  2. The sphere growth assay should be better quantified. Why using area but not numbers? And representative images need to be provided in the main figures.
  3. Figure 7 is not helpful and over-interpreted. The authors should focus on the results generated in the current study. 

Reviewer 2 Report

In this manuscript, Minami and colleagues examined if lactate can be used as alternative carbon source by glioma stem cells in the context of glucose restriction. The authors studied the role of lactate in reprogramming energy and lipid metabolism of glioma stem cells in glucose- deprived conditions. They found that lactate can support oxygen consumption of oxidative glioma stem cells, contribute to the generation of metabolic intermediates of the TCA cycle and modulate the redox status of the cells as well as reprogram fatty acid synthesis, all of which supporting tumor growth and aggressiveness. The study is very relevant to the filed of brain cancer and further illustrates the level of metabolic heterogeneity and plasticity present in this disease and further support the need to understand the metabolic regulations taking place in these tumors with the potential to identify metabolic vulnerabilities to be exploited therapeutically.

The manuscript is well written and the experiments are conducted in a careful and thoughtful manner, the data included are rich, and support most conclusions well. Overall, the report is strong and I would recommend acceptance following adequate responses to the following points:

  • Figure 1: what is the lactate production of GSCA? In addition to sphere area did the authors observe changes in sphere forming frequency in response to the different concentration of lactate?

  • A significant portion of the data presented are normalized and relative to controls. It might be more relevant to show the raw values wherever possible such as for the sphere areas.

  • Figure 3F: please present representative flow plots

  • LC/MS raw data should be included as supplemental information.

  • Figure 6: the authors indicated that lactate sustains tumorigenicity of GSCs however they did not directly measure and compared tumorigenicity, which would require limiting dilution transplantation assay, but rather investigated rate of tumor growth or tumor aggressiveness. The author should revise their statements about tumorigenicity accordingly.

Reviewer 3 Report

Minami et al. present a well written paper on how lactate can fuel murine glioma stem cells that rely on oxidative phosphorylation. It is demonstrated that lactate does not only contribute to mitochondrial ATP production, but also serves as carbon source for anabolic processes, such as lipid synthesis, and reduces cell toxic ROS levels. In this model, lactate was capable of sustaining tumorigenesis in low glucose conditions.

I very much enjoyed reading this manuscript, and recommend publication in the current form.

Author Response

We thank the reviewer for the critical reading and the positive feedback for our manuscript.

Round 2

Reviewer 1 Report

The authors have addressed all my comments and suggestions.